# Maternal Singing but Not Speech Enhances Vagal Activity in Preterm Infants during Hospitalization: Preliminary Results

**DOI:** 10.3390/children9020140

**Published:** 2022-01-21

**Authors:** Manuela Filippa, Mimma Nardelli, Elisa Della Casa, Alberto Berardi, Odoardo Picciolini, Sara Meloni, Clara Lunardi, Alessandra Cecchi, Alessandra Sansavini, Luigi Corvaglia, Enzo Pasquale Scilingo, Fabrizio Ferrari

**Affiliations:** 1Department of Psychology and Educational Sciences, University of Geneva, 24 Rue General Dufour, 1211 Geneva, Switzerland; 2Department of Social Sciences, University of Valle d’Aosta, Str. Cappuccini 2, 11100 Aosta, Italy; 3Bioengineering and Robotics Research Centre E. Piaggio, 56122 Pisa, Italy; mimma.nardelli@ing.unipi.it (M.N.); e.scilingo@ing.unipi.it (E.P.S.); 4Dipartimento di Ingegneria dell’Informazione, University of Pisa, 56122 Pisa, Italy; 5Neonatal Intensive Care Unit, Women’s and Children’s Health Department, University Hospital of Modena, 41124 Modena, Italy; dellacasa.elisa@policlinico.mo.it; 6Department of Medical and Surgical Sciences of Mother, Children and Adults, University Hospital of Modena, 41124 Modena, Italy; alberto.berardi@unimore.it (A.B.); fabrizio.ferrari@unimore.it (F.F.); 7Pediatric Physical Medicine & Rehabilitation Unit, IRCCS Ca’ Granda Ospedale Maggiore Policlinico, Via Francesco Sforza 35, 20122 Milan, Italy; odoardo.picciolini@policlinico.mi.it (O.P.); sarachiara.meloni@gmail.com (S.M.); 8Department of Neurosciences, Psychology, Drug Research and Child Health, Careggi University Hospital of Florence, 50139 Florence, Italy; clara.lunardi@unifi.it; 9Division of Neonatology, Careggi University Hospital, School of Medicine, University of Florence, 50134 Florence, Italy; ale.cecchi76@gmail.com; 10Department of Psychology “Renzo Canestrari”, University of Bologna, Viale Berti Pichat 5, 40127 Bologna, Italy; alessandra.sansavini@unibo.it; 11Neonatal Intensive Care Unit, IRCCS AOU Bologna, Via Massarenti 9, 40138 Bologna, Italy; luigi.corvaglia@unibo.it; 12Department of Medical and Surgical Sciences, University of Bologna, Via Massarenti 9, 40138 Bologna, Italy

**Keywords:** early vocal contact, heart rate variability, maternal voice, preterm infants

## Abstract

Background: Early parental interventions in the Neonatal Intensive Care Units (NICUs) have beneficial effects on preterm infants’ short and long-term outcomes. The aim of this study was to investigate the effects of Early Vocal Contact (EVC)—singing and speaking—on preterm infants’ vagal activity and autonomic nervous system (ANS) maturation. Methods: In this multi-center randomized clinical trial, twenty-four stable preterm infants, born at 25–32 weeks gestational age, were randomized to either the EVC group or control group, where mothers did not interact with the babies but observed their behavior. Heart Rate Variability (HRV) was acquired before intervention (pre-condition), during vocal contact, and after it (post condition). Results: No significant effect of the vocal contact, singing and speaking, was found in HRV when the intervention group was compared to the control group. However, a significant difference between the singing and the pre and post conditions, respectively, preceding and following the singing intervention, was found in the Low and High Frequency power nu, and in the low/high frequency features (*p* = 0.037). By contrast, no significant effect of the speaking was found. Conclusions: Maternal singing, but not speaking, enhances preterm infants’ vagal activity in the short-term, thus improving the ANS stability. Future analyses will investigate the effect of enhanced vagal activity on short and long-term developmental outcomes of preterm infants in the NICU.

## 1. Introduction

Several studies have reported an association between prematurity and infants’ ANS maturation during development [1,2]. One of the best predictors of ANS maturation in the neonatal period is the heart rate variability (HRV), measured as the variation in beat-to-beat (R-R) intervals between two consecutive heart beats [3]. Even if heart rate levels are similar, a healthy full-term neonate has a better beat-to-beat variability than a high-risk preterm neonate [4]. In neonates, improved maturation of the ANS promotes survival during critical periods of hospitalization. The ANS continuously matures from the last half of gestation through the neonatal period [5], and it reflects, through HRV measurements, the gradual activation of the parasympathetic tone, which increases near the term age [6]. The developing brain in the perinatal period assures that higher cortical processes can integrate autonomic control, with a consequently increased influence of the parasympathetic system on infants’ regulation activity [7].

In a relational perspective, HRV provides a physiological stand for developing the infant’s abilities to interact with caregivers and objects during development [8].

In low-risk samples of premature infants, such as the population involved in the present study, the association between duration of preterm extrauterine maturation and ANS development was not significant before the term-corrected age [9]. Thus, in stable populations of preterm infants, with no additional comorbidities other than prematurity, we can presume that fluctuations in the HR could be mainly determined by environmental intervening factors.

Indeed, the relational environment is fundamental for developing the infants’ reactions to social and environmental stimuli, which are mediated by a correct equilibrium between the sympathetic and parasympathetic systems [10]. This equilibrium allows for the emerging of a contingent response to external salient sensory stimuli, which, in turn, constitutes the basis for later co-regulatory experiences [11].

Early kangaroo-care enhanced preterm infants’ autonomic nervous system (ANS) development, as measured by its vagal activity [12]: infants displayed an improved vagal tone maturation between 32 and 37 weeks of gestational age, with a concomitant improvement in state organization. Moreover, autonomic regulation of preterm infants is enhanced by Family Nurture Intervention [13]. Finally, exposing preterm infants to their mothers’ voices in the NICU affected infants’ physiological regulation by increasing oxygen saturation, decreasing episodic apnea and bradycardia, and positively impacting their stability [14,15,16,17,18]. More specifically, live maternal singing during skin-to-skin contact induced significant change in low frequencies (LF) and high frequencies (HF) in the HR. Lower LF/HF ratio was observed during the vocal intervention and recovery phases, compared with standard kangaroo-care and baseline [19].

### Early Vocal Contact (EVC) in the NICU

EVC between parent and premature infant has positive short-term effects on the infant’s physiological stability and behavior [20,21]. When the mother speaks, the newborn opens its eyes and tends to transition towards a quiet wakefulness; when the mother sings, the initial infants’ behavioral state tends to remain stable [20]. During maternal speech and singing, compared to the absence of vocal contact, the preterm infants increase self-touch gestures and eye opening [21]. Finally, during routine, painful, procedures, maternal speech decreases pain scores in preterm infants and increases their endogenous oxytocin release, with potential analgesic effects [22]. The same tendency is evidenced during singing, but the results are only marginally significant [22]. The results mentioned so far suggest that maternal singing and speech could have differential effects on hospitalized premature infants.

In the present study, we hypothesized that the live maternal voice, both speaking and singing, would increase HRV during EVC.

This study is part of a larger study investigating the short and long-term effects of EVC on preterm infants [23].

## 2. Materials and Methods

### 2.1. Design

A four-site randomized controlled trial has been conducted to investigate the short-term physiological effects of EVC on preterm infants’ vagal activity, using Heart Rate Variability analysis. Infants were recruited from four university hospitals: Careggi University Hospital (Florence), NICU Fondazione IRCCS Ca’ Granda Ospedale Maggiore Policlinico (Milan), Modena University Hospital (Modena), and Bologna University IRCCS Hospital (Bologna).

### 2.2. Participants

Twenty-four preterm infants, born at 25–32 weeks and 6 days GA, were recruited from the four centers. Twelve subjects were assigned to the control group and 12 to the experimental group, randomizing infants by gender and GA.

Due to centralized randomization, children were included in all centers involved, both for the experimental and control group. Descriptions of the involved population are summarized in Table 1.

After obtaining informed consent, families were enrolled. The inclusion criteria for the newborns were: GA between 25 + 0 and 32 + 6 weeks at birth; Apgar score >7 at 10 min; birth weight between 3rd and 97th percentiles; birth cranial circumference greater than 10th percentile; periventricular leukomalacia grade 1; and intraventricular hemorrhage grade 1–2. Preterm newborn infants were excluded in presence of sepsis, congenital malformations and/or genetic abnormalities. Mothers were excluded if they had clear depressive symptoms, drug abuse, and/or age below 18 years.

### 2.3. Intervention

Mothers were asked to speak and sing to their infants over a 10 min period for each type of intervention (20 min in total), three times per week, for 2 weeks, midway between the feeding cycles, more than 4 h after the last medical exam. EVC began when the newborns were in an active sleep state, in calm awake state, or in active awake state, but not in deep sleep or crying.

The order of the two vocalizations was counterbalanced for the same subject. If the patient on day 1 started with singing then, on day 2, the first vocalization was speech (See Figure 1). The mothers were free to choose which vocalization to start with on day 1.

During the EVC, infants were either in their individual incubators or open cribs. Mothers in the control group were encouraged to spend the same amount of time as that of mothers in the intervention group, observing their infants’ spontaneous behavior, with the subsequent compilation of an observation grid developed ad hoc, according to a few indicators drawn from The Neonatal Behavioral Assessment Scale.

### 2.4. Main Outcome and Outcome Measures

HRV was measured through the acquisition of the Photoplethysmographic (PPG) signal, via sensors placed on the infant’s foot by routine medical devices. Subsequently, the PPG signals were automatically recorded in each center by the same IxTrend Software (Ixellence GmbH, Wildau, Germany). PPG signals were collected over the intervention period, as well as 20 min before and after the mother’s intervention. In the control group, the same signal was collected 20 min before, in absence of the mother, 20 min during the silent maternal presence, and 20 min after it. Note that, in the control group, mothers simply observed their infant, without vocalizing or touching the infant.

### 2.5. Data Analysis

#### 2.5.1. Photoplethysmographic (PPG) Signal Processing

PPG signals acquired from 24 subjects were analyzed, including 12 subjects from the control group and 12 from the experimental group. In the experimental group, singing and speaking were investigated. PPG signal was acquired before intervention (pre-condition), during vocal intervention, and after it (post-condition). The same signal was acquired for the control group, as well as before and after it.

The sampling frequency of PPG signals was 125 Hz. A band-pass filter (cut-off frequencies of 0.5–10 Hz) was applied to each PPG series, and the pulse-to-pulse time series (PP series) was obtained using an algorithm based on a low-pass differentiator filter [24]. The smoothness priors’ approach was applied to each pulse-to-pulse series as detrending method [25].

For each subject, three time-windows, lasting five minutes each, were considered for further analyses: one time-window was extracted from the pre-condition, preceding the vocal contact, while the second window corresponded to the first five minutes of intervention—singing or speaking—, and the third time-window was extracted from the post-condition, following the vocal contact. The same time-windows were extracted for the control group. After the pre-processing, a total of 36 time-windows, lasting five minutes each, for the control, and 34 time-windows for the intervention groups were included in the analyses. Each subject experienced a minimum of 2 and a maximum of 6 (Mean = 3) repeated sessions.

#### 2.5.2. Heart Rate Variability Feature Extraction

PP series were converted into HRV signals, using a cubic spline interpolation to resample each series at a rate of 4 Hz. In the next step, the power spectral density of each of the HRV signals was analyzed using a time-varying parametric spectrum estimation based on an autoregressive model [26]. Two main bandwidths were investigated, i.e., a low frequency band (between 0.02 and 0.2 Hz) and a high frequency band (between 0.2 and 1.5 Hz) [27]. Considering each bandwidth, we computed the values of five features:-Low Frequency (LF) power and High Frequency (HF) power: the absolute power values;-LF power nu and HF power nu: the related values normalized to the sum of the LF power and HF power (LF power nu = LF power/(LF power + HF power) and HF power nu = HF power/(LF power + HF power));-LF/HF: the ratio between LF power and HF power.

Considering that more than one recording session was performed for each subject, the median value of the features calculated on all available sessions was used in the statistical analyses (see Section 2.5.3).

#### 2.5.3. Statistical Analysis

Two different statistical analyses were applied to HRV data. An intra-group statistical analysis aimed to study the differences in HRV dynamics among the three Time Conditions (pre, during, and post) in each group (intervention and control). Given the non-Gaussianity of data distributions, we applied a non-parametric Friedman statistical test to compare the values of each HRV feature in the three conditions. After this, we compared the feature values found for each pair of time conditions using a two-tailed Wilcoxon signed-rank. Bonferroni’s test for post hoc analysis was applied in order to test the statistical differences between each pair of conditions [28].

The Kruskal–Wallis non-parametric statistical test was used for the inter-group statistical analysis. We subtracted the values of the HRV features during the pre-condition from the values of the HRV features during singing, speaking, and the control group. In the next step, we applied the Kruskal–Wallis test to investigate the differences among the groups. The Mann–Whitney test was used to compare each pair of vocal interventions (singing vs. speaking). Likewise, the same test was used to compare singing and speaking in the control group. Additionally, in this case, Bonferroni’s test was used for post hoc analysis of multiple comparisons.

## 3. Results

There was no significant effect of the intervention, singing and speaking, found in the HRV when compared to the control group, after the application of Kruskal–Wallis test (LF power: *p* = 0.977; HF power: *p* = 0.893; LF nu: *p* = 0.224; HF nu: *p* = 0.224; LF/HF: *p* = 0.3436). However, significant differences were found in the HRV features in the intra-group analysis, for the time conditions, when we compared the pre, singing and post conditions (*p* = 0.013). Specifically, after the Friedman test a *p*-value < 0.05 was assigned to three features: LF power nu, HF power nu, and LF/HF. Bonferroni’s test for post hoc analysis revealed a significant difference between the singing condition and the pre and post conditions (LF power nu and HF power nu: *p* = 0.015 and 0.037, LF/HF: *p* = 0.021 and *p* = 0.028). Figure 2, Figure 3 and Figure 4 show the median and median absolute values (MAD) of LF power nu, HF power nu, and LF/HF features, for each of the three time conditions for control, singing, and speaking groups. No significant differences were found in the three time conditions—pre, during, and post—for the speaking group. Likewise, no significant differences were found in the three time conditions—pre, during, and post—for the control group.

## 4. Discussion

The preliminary results showed that preterm infants’ HRV is modulated in the short-term by the maternal live singing. An assessment of the short-term HRV, as in the present work, offers important evidence about maturation and dynamical equilibrium of the ANS in newborns. More specifically, the median and median absolute values of HF were significantly higher during singing than in the pre and post condition, in absence of the mother’s voice, indicating an improved stability of the ANS during the singing interaction. HF component of the HRV is correlated to parasympathetic activity, and the related spectral activity in preterm infants is lower than in term infants; this is mainly due to their ANS immaturity [29]. Thus, higher values of HF indicate an increase in maturation of parasympathetic activity, and, indirectly, can be an index of the ANS maturation in preterm infants [30]. During maternal singing, both LF and LF/HF features of the HRV were significantly lower than in the pre and post condition: these two components reflect both sympathetic and vagal dynamics [31,32], and the ratio of LF to HF can be interpreted as an index of physiological arousal. The dominant activity of the LF is a typical consequence of preterm birth, demonstrating a dominance of the sympathetic system over the parasympathetic. In our study, preterm infants, exposed to maternal singing, show a decrease of the LF spectral components, with a concomitant increase of the HF. This concomitant fluctuation expresses, again, an improved balance between these two components and an increase in the infant’s vagal activity.

We did not find the same effect with speech. As previously mentioned, maternal speech has activating effects on the premature infant, and it induces active waking states, as opposed to singing, which tends to stabilize the infant’s state [20]. Furthermore, in a recent study, we have shown that maternal speech has a significant effect on the oxytocin release of the premature infant during a heel prick procedure, whereas these effects are marginally significant for singing. Finally, only maternal speech significantly modulates the infant’s pain, reducing its facial and physiological manifestations in infants, whereas this effect was not found for maternal singing [22]. We can therefore conclude that, during early vocal contact, maternal singing and speech seem to create different effects on the behavior, physiology and, more generally, on the ANS of premature infants. Maternal singing, in fact, with its repetitive structure, homogeneous with respect to a predefined melodic line and rhythmicity, has a significant impact on HRV and therefore on the vagal activity of the newborn [19]. Speech, on the other hand, contains the peculiarity of modulating on the behavioral signs of the newborn [33]. It does not have a predefined structure like singing, but it responds to the immediate changes in the infant’s state, and it acts on its endocrine system, with an increase in oxytocin. Thus, maternal singing and speech seem to have different functions in the early vocal communication between parents and infants. Both forms of communication have evolved with humans over millennia, both have characteristics that distinguish them from adult-directed speech and singing, both modulate to the age and development of the child, and both convey emotional contents to which infants are sensitive [34]. However, they also show significant differences. In a naturalistic context, infant-directed singing is more predictable on the basis of melodic and rhythm repetitions [35], as opposed to speech, and its motivational and structuring properties facilitate learning. Most importantly, singing is used by adults to modulate the infants’ state of arousal, sometimes to calm them with lullabies, and sometimes to activate them with play songs [36].

Moreover, the same differences between the pre, during, and post maternal singing were not found in the control group, in which the mothers were present but merely observed the infant’s behavior. Therefore, the silent presence of the mothers next to the incubator is not sufficient by itself to modulate the infant’s cardiac responses and stimulate its vagal activity.

Finally, no significant differences were found between the intervention and control groups. As a non-significant trend is evidenced in the group comparisons, it is possible that an increase in the number of subjects will contribute to deeper statistical analyses concerning the differences between the groups under examination, thus, raising the statistical power of the study.

## 5. Conclusions

Preterm infant-directed singing has modulatory effects on infants’ HRV, increasing the vagal activity in newborns. An enhanced autonomic regulation facilitates early nurturing interactions between preterm infants and their parents [13]. These preliminary results, even if assessed in a small population, are encouraging, and they provide a foundation for future analyses. The primary outcome, HRV, will therefore be analyzed on a larger sample of infants during early vocal contact. In addition, the physiological data will be correlated with the degree of maternal stress and, above all, with the neuromotor maturation assessed in general movements [37]. As a final step of our future analyses, neonatal HRV data, in response to contingent vocal contact, will be correlated with the language skills of infants involved in early intervention.

## 6. Limitations and Future Works

HRV series of preterm infants were extracted from PPG signals. Choosing to acquire PPG signals instead of ECG, which is the gold standard for HRV extraction, has several advantages. Specifically, PPG sensors are less obtrusive than ECG acquisition systems, and they can be easily integrated into wearable devices.

However, we are aware of the less accurate peak detection on PPG signals compared to ECG signals [38]. Future studies will be addressed towards the comparison of the results obtained through the analyses of PPG and ECG signals, and, additionally, towards the application of multivariate indexes for a comprehensive assessment of cardiovascular dynamics [39]. The number of subjects involved in the study will be increased in order to raise the statistical power of the intra-group and inter-group analyses.

## Figures and Tables

**Figure 1 children-09-00140-f001:**
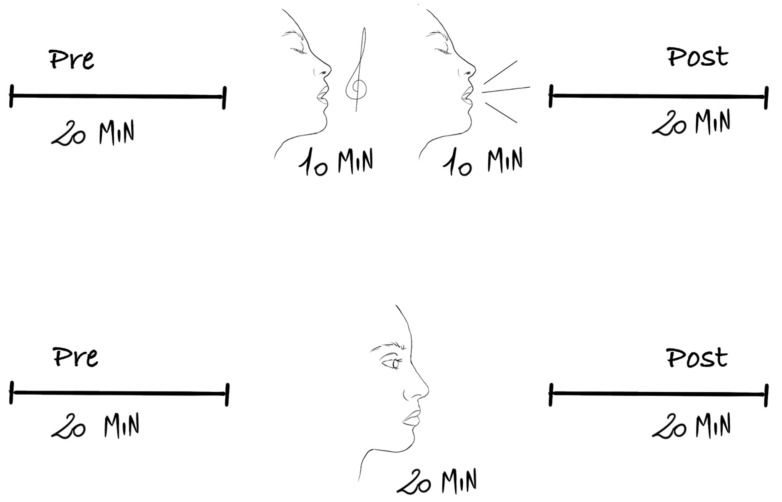
Illustrations of the intervention (top) and the control (bottom) groups. The order of singing and speaking was reversed on the following day.

**Figure 2 children-09-00140-f002:**
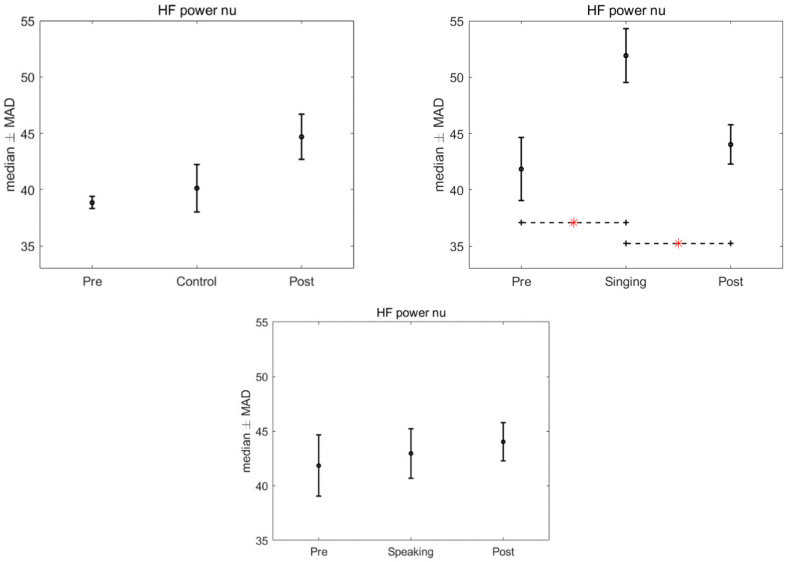
Median and MAD of HF power nu for the three time conditions related to control, singing, and speaking groups. The asterisk indicates the sessions that resulted significantly different after we used Bonferroni’s test for post hoc analysis.

**Figure 3 children-09-00140-f003:**
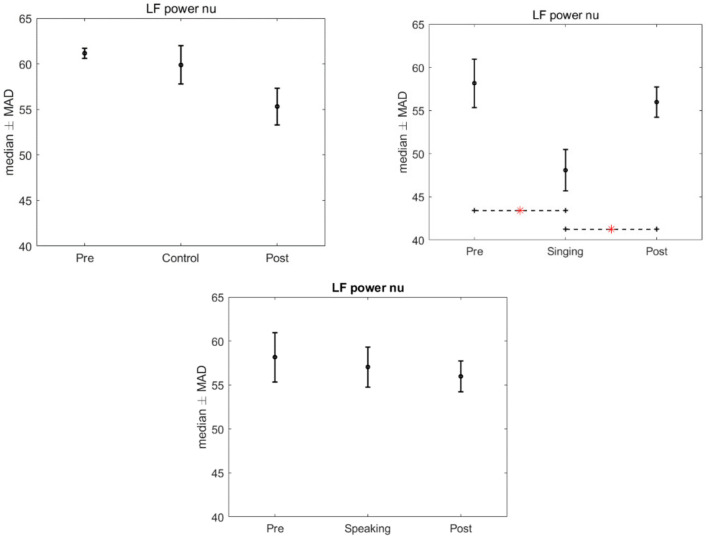
Median and MAD of LF power nu for the different time conditions related to control, singing, and speaking groups. The asterisk indicates the significant comparisons observed after we used Bonferroni’s test for post hoc analysis.

**Figure 4 children-09-00140-f004:**
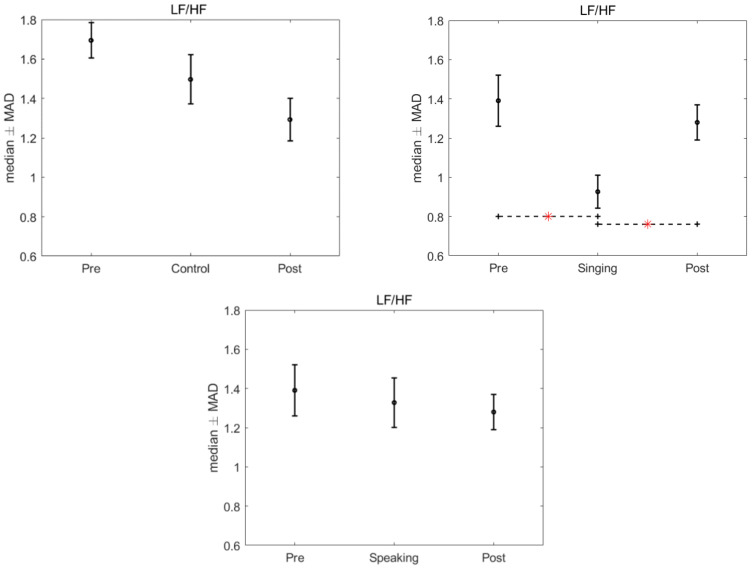
Median and MAD of LF/HF for the three time conditions related to the control, singing and speaking groups. The asterisk indicates the significant comparisons observed after we used Bonferroni’s test for post hoc analysis.

**Table 1 children-09-00140-t001:** Infants and mothers’ demographics.

Characteristics	Intervention Group (*n* = 12) Mean (SD)	Control Group (*n* = 12) Mean (SD)
Gestational age at birth (weeks)	29.1 (2.4)	29.3 (1.5)
Birthweight (g)	1226 (367)	1204 (235)
Apgar score at 5 min	7.1 (1.3)	6.4 (2.3)
Apgar score at 7 min	8.1 (1.1)	7.9 (1.0)
Gestational age at test (weeks)	34.1 (1.3)	34.2 (1.7)
Post-natal age at test (days)	33.8 (19.2)	34.7 (15.8)
Weight at test (g)	1767.8 (258)	1603.8 (308)
Maternal citizenship	Italian (80%)	Italian (70%)
Primiparous	Yes (72%)	Yes (60%)
Mother’s age	35.6 (3.2)	37.7 (6.5)

## Data Availability

All data are available upon request.

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
