# Peer review of "Maternal Singing but Not Speech Enhances Vagal Activity in Preterm Infants during Hospitalization: Preliminary Results"

_children, 2022, doi:10.3390/children9020140_

Round 1

Reviewer 1 Report

Review Children 1487

Maternal singing but not speech enhances vagal activity in pre- 2 term infants during
hospitalization in the Neonatal Intensive Care Unit: preliminary results
The authors investigated the effects on EVC (Early Vocal Contact) on preterm infants. More
specifically, they hypothesized that the live maternal voice, both speaking and singing, would
increase HRV (Heart Rate Variability) during EVC.
The paper is very interesting, and I believe it will interest many readers from various fields,
such as pediatrics, cognitive science, developmental science and neuroscience. I do,
however, have some concerns, mainly methodological points that I hope will help readers to
better understand the feasibility of the study. I offer my constructive comments, which I
hope that they are useful to the authors.
The introductory chapter is very interesting and well-constructed. However, I would urge the
authors to be a little more careful in the conclusions of the studies cited. For example, p2
line 100, does EVC significantly increase self-touch gestures? If the answer is no, it should be
stated that it is not significant and no conclusions should be drawn about the behavior of
preterm infants. Similarly, on p3 line 105, are the potential analgesic effects significantly
revealed? If the answer is no, they should not be mentioned.
In the chapter Materials and Methods, I have several considerations:
1. What is the distribution of the groups (experimental and control) according to the 4
university hospitals? Are the experimental groups in 2 centers and the control groups in the
other 2 centers or does each center have both experimental and control children?
2. P3, The demographic characteristics of table 1 should be given for the experimental
group and for the control group in order to judge the homogeneity of the demographic
characteristics between the two groups of preterm children.
3. P4, line 149 and 150: If I understand correctly the sentence "the order of the two
vocalizations, speaking and singing, was reversed in the following intervention", in the
experimental group the mother speaks first during 10 minutes and then she sings during 10
minutes. Is there no counterbalanced order between the subjects nor for the same subjects
during the 6 times in the 2 weeks?
4. P4, line 152: the word condition is confusing. It is experimental (intervention) group
and control group, but not condition. What intervention and control conditions means?
5. P4, line 152: if the order of the two interventions conditions (speaking and singing) is
reversed, why not present on figure 1 speaking first and singing second, that would be less
confusing
6. P4, line 154: The fact that the child is in an incubator provides different stimuli from
those received by children in open cribs: auditory stimuli do not reach the ears of the
premature child under the same conditions (resonance of auditory stimuli inside the incubator). The same is true for visual stimuli, as the plexiglass of the incubator acts as a barrier for visual stimuli.
6. P4, line 169: control condition mothers »: again, is it the control group or the control
condition?
7. P5, line 175: How long did the acquisition last before, during and after the
intervention? Was this measure controlled in the same way in all university hospitals and for
all groups (experimental and control)?
8. P5, line 181: Is it the same equipment to record the signal in the 4 different university
hospitals ?
9. P5, line 188: I dont understand the numbers 36 and 34. Please justify, it would be more easy to understand.
In discussion section,
P7, line255: It is not vocal interaction but it is singing interaction.

Author Response

Dear Reviewer 1,

We would like to thank you for your precious suggestions, which have helped us to improve the clarity of the manuscript.

You’ll find in attachment the answers to your comments.

Best regards

Reviewer 2 Report

The paper addresses the influence of maternal vocal contact on the vagal activity in preterm infants. It is a relevant topic showing that small changes in daily NICU practice could have impact on the outcome of the infants.

In general the review is easy to read, giving a good overview of preceding studies.

The Introduction is well written, leading to a clear research question and setup.

Methods:

  • The study design with an RCT is well thought of and a good setup to investigate effects. I would then advice to include in Table 1 for the groups separately the Mean (SD).
  • In 2.3 the part on sleep is described twice.
  • How accurately is the timing of ‘start of singing’ and ‘end of singing’ session timed? Only the first five minutes of a singing session is used: why? And if there may be inaccuracies in the timing of the start of the session, would then 5 minute segment in the middle of the session be better?
  • The main concern with this paper is the technical description in 2.4, which is very confusing as first is mentioned that ECG is measured and used for HRV, but in 2.5 the PPG signal is used for HRV analyses, at a sample rate of 125 Hz. The paper does not contain a limitations section, where at least the limitations in the less accurate PPG signal and the less accurate peak detection on PPG signals should have been mentioned.
  • A mean of 3 sessions per infant is reported: what was the range? Could one infant contribute much more to the mean than others?

Results:

  • Why are the results of the control group not shown in Figure 2 as well? It is now unclear why the median that differs significantly between singing and pre does not differ between singing group and control group. You would have expected this, as the control group does not show pre & ‘intervention’ differences. It that because the control group shows different values for HF and LF? By showing in the same graph, this may help the reader to understand.

Discussion:

  • Why is there no discussion on the RCT method and that results between intervention group and control do not show differences.
  • A section on limitations of the study misses.
  • The statement that there is a potential long-term effect is not proven by this study, I would suggest to make it less strong.

Author Response

Dear Reviewer 2,

We wish to thank you for carefully reading our manuscript and for all the pertinent comments.

You’ll find in the following lines the answers to your questions and comments.

Best regards

Round 2

Reviewer 2 Report

The revised manuscript has improved significantly. It is now a coherent story and the data is now shown well.